# Psychometric Properties of the Spanish Version of the Work Group Emotional Intelligence Profile Short Version (WEIP-S) in a Sample of Spanish Federated Coaches

**DOI:** 10.3390/ijerph192114371

**Published:** 2022-11-03

**Authors:** Miriam Granado-Peinado, Carlos Marchena-Giráldez, Inés Martínez-Infiesta, Jorge Acebes-Sánchez

**Affiliations:** 1Faculty of Education and Psychology, Universidad Francisco de Vitoria (UFV), Ctra. Pozuelo-Majadahonda Km 1.800, 28223 Pozuelo de Alarcón, Spain; 2Faculty of Health Sciences, Universidad Francisco de Vitoria (UFV), Pozuelo de Alarcón, 28223 Madrid, Spain

**Keywords:** emotional intelligence, WEIP-S, validation, coaches, sport

## Abstract

Emotional intelligence has been a topic of great interest to researchers in many different areas as it is associated with mental, psychosomatic, and physical health. In the sports context, it is a significant variable that can play an important role in improving the team’s performance. Although there are numerous tools to assess emotional intelligence, few of them have been validated explicitly in a sports sample, and even fewer have had coaches as a target population. Therefore, this study aimed to validate the Spanish version of the work group emotional intelligence profile short version (WEIP-S) in a sample of Spanish federated coaches. The results confirm that this instrument presents good psychometric properties to measure the emotional intelligence of sports coaches. The original four-factor model (awareness of one’s own emotions, management of one’s own emotions, awareness of others’ emotions, and management of others’ emotions) shows good reliability and convergent validity for all four factors except for the management of one’s own emotions. These findings suggest that it is possible to measure the emotional intelligence of coaches and offer the opportunity to continue investigating the relevance of constructing specific scales to measure this construct in the sports context.

## 1. Introduction

Emotional intelligence (EI) is a term first introduced by the authors Mayer and Salovey [1]. It was defined as “an individual’s capability to perceive, use, understand, and manage emotions”. EI has been a matter of great interest for investigators in many different areas, concluding that increased EI is associated with mental, psychosomatic, and physical health results [2]. Some studies show that the higher the physical activity, the higher the EI [3,4]. Specifically, in the sports context, different studies showed that EI is related to sports performance [5].

In the systematic review of 36 articles on EI in a sport or exercise context, Laborde et al. [4] found not only that it was beneficial for athletes to have higher levels of EI but also that EI could be taught. They also found that there was a lack of research on the EI of coaches, officials, and spectators.

Coaching requires soft skills such as EI, motivation, inspiration, or conflict management [6]. Thus, coaches play an important role in creating the emotional atmosphere of youth sports [7]. This emotional atmosphere can be facilitated by the psychosocial features of various coaches, such as leadership style [8], goal orientation [9], expectations [10], and coach behaviour in a competitive environment. A coach who is unaware of his own emotions is unable to regulate them correctly and, eventually, needs to understand their emotions for the well-being of his players [11]. Indeed, there is evidence that coaches’ in-game behaviour affects their relationship with athletes and their psychological performance [12]. Therefore, the impact of the coaches’ behaviour on the athletes’ performance and well-being has received significant attention in youth sports. On the other hand, Teques et al. [13] found that coaches who feel competent in regulating their emotions perceived that they could motivate and build the character of their athletes. This insight has an impact on their positive verbal reactions in response to athletes’ performances.

Campo et al. [14] sought to work with athletes to improve their emotional skills. They explored the impact of EI training programs that matched elite team sports, delivered by three EI coaches: the team coach, the team physiotherapist, and a sport psychology expert. The results of the study showed that the type of emotional competencies developed depended on the status of the EI trainers. These results highlight the appropriateness of a group-based approach. This is consistent with some of the research conducted on this topic (e.g., [15]) in which leaders with low levels of EI had greater difficulties in leading. On the other hand, coaches who can assess their own emotions will be more sensitive in regulating them according to the situation, enabling them to fulfil their role as a coach [16].

On another subject, multiple investigations have validated EI measurement instruments for sports contexts [17]. They mostly used the SSRI, trait meta-mood scale (TMMS-24) [18], and bar-on emotional quotient inventory (EQ-I) [19] as theoretical base models. Marchena-Giráldez et al. [20] validated the Spanish version of the work group emotional intelligence profile short version (WEIP-S) in the sports context (athletes). They concluded that EI is not merely an individual construct but rather something that affects all team members (teammates, coaches, staff). In this context, this study aims to validate the Spanish version of the WEIP-S questionnaire for the measurement of EI in coaches and to analyse the psychometric properties of this tool in a sample of Spanish federated coaches in terms of validity and reliability.

## 2. Materials and Methods

### 2.1. Participants

The sample in this research consisted of Spanish men and women federated coaches. The data collected by the Ministry of Culture and Sport were used to determine the sample size. The data for 2019 show that the number of coaches trained in the strictly federative field during 2017 was 8109, with a total of 404 courses given. The results by gender indicate that 75.3% of the trained coaches correspond to men and 24.7% to women [21]. Thus, the minimum number of participants in our study is 150 (confidence interval = 95%; margin of error = 5%; population proportion = 8.27%). Here, we evaluate 161 coaches from different sports styles. The sample is for convenience and non-random. The average age is 36.69 years (SD = 10.31). Table 1 summarises the specific features of the sample. The inclusion criteria were the following: being of legal age, coaching a sport at a local, provincial, national, or international competition level, and being a coach at the federated level in the last two years or currently being a coach at the federated level. The exclusion criterion was not being a coach at the federated level.

### 2.2. Measures

Participants completed a sociodemographic questionnaire created ad hoc to include data on several variables (age, sex, sport modality, years, and frequency in which they coach their sports team, etc.). They also filled out the standardised measures below to analyse the convergent and concurrent validity of the WEIPS-S:Workgroup Emotional Intelligence short version (WEIP-S) [22], in the Spanish version by Lopez-Zafra et al. [23]. The questionnaire was composed of 16 items to evaluate EI in the workgroup setting. Answers followed a Likert scale that ranges from 1 (strongly disagree) to 7 (strongly agree) and is categorised into four dimensions (with four items in each dimension): awareness of one’s own emotions (e.g., item 2: I can explain my emotions to other team members), management of one’s own emotions (e.g., item 6: when I am frustrated with a team member, I can overcome my frustration), awareness of others’ emotions (e.g., item 10: I am able to accurately describe how team members are feeling) and management of others’ emotions (e.g., item 14: I am able to encourage team members when they are feeling down). Internal consistency values varied between good (α = 0.71) and excellent in all dimensions (α = 0.91).Trait Meta Mood Scale (TMMS-24) [24] in the Spanish version of Fernández-Berrocal et al. [18] to measure convergent validity. This self-reported assessment consists of 24 items to measure the individual’s EI. Responses are presented on a Likert scale ranging from 1 (never) to 5 (very often) and are organised into three dimensions (with eight items in each dimension): emotional attention, emotional clarity, and emotional repair. The Spanish validation revealed good values of internal consistency in all subscales (above α = 0.85), as well as temporal stability (from r = 0.60 to r = 0.83). This scale has also been validated in the sports context, showing appropriate reliability and construct validity [25].Perceived Stress Scale (PSS) [26] in the Spanish version of Remor [27] to measure concurrent validity. It is composed of a unidimensional scale of 14 items to assess the level of self-perceived stress in the last month (e.g., item 2: In the last month, how often have you felt unable to control the important things in your life?) Responses are given following a Likert scale ranging from 0 (never) to 4 (very often). The evaluation of psychometric properties revealed good values for internal consistency (α = 0.81) and test—retest temporal stability (r = 0.73).

### 2.3. Procedure

Participation was requested by email to sports clubs in Spain and by snowball sampling. The data collection was from 6 April to 27 April 2021. Having reviewed the overall background information on the aim of the survey, the participants completed an informed consent form to indicate their acceptance to participate in the study. Under no circumstance did participants benefit in any way; such participation was entirely voluntary. Information was obtained via the Google Forms platform. No personal data were requested from participants to ensure confidentiality. It took about 20 min to complete the survey. We collected sociodemographic information, and the WEIP-S, the TMMS-24, and the PSS were administered. The study fully complied with the Declaration of Helsinki and was endorsed by the Research Ethics Committee of the Universidad Francisco de Vitoria (16/2020).

### 2.4. Data Analysis

R software (https://www.r-project.org/ (accessed on 22 July 2022) was used to compute data analysis. Firstly, we analysed descriptive analysis for all the measures by univariate and multivariate tests with Kolmogorov-Smirnov and Mardia tests. Secondly, we compute several confirmatory analyses for four models: (a) the unifactorial model; (b) the model proposed by the authors composed of 4 correlated factors: awareness of own emotions (AE), management of own emotions (ME), awareness of others’ emotions (AOE), and management of others’ emotions (MOE); (c) the bifactor model, where the correlation between the general emotional intelligence factor and the specific factors are constrained to zero; and (d) a hieratic model where the key four factors proposed by the authors can be the group where they are defined as a general factor.

We assessed model fit using RMSEA (root mean square error of approximation) with a 90% confidence interval, the CFI (comparative fit index), the TLI (Tucker-Lewis Index), and the SRMR (standardised root mean square residual). Values > 0.95 for CFI and TLI and values < 0.06 for RMSEA and SRMR indicate a good model fit. Additionally, AIC (Akaike information criterion) and BIC (Bayesian information criterion) were analysed to compare the models. To interpret these statistics, lower values indicate a better model.

Finally, some Pearson correlation coefficients were explored to analyse nomological validity by the association of the WEIP-S with related constructs such as the Trait-Meta Mood Scale (TMMS-24) and the Perceived Stress Scale (PSS). Higher correlations are expected between the WEIP-S factors and the TMMS-24 factors, while lower correlations are expected between the WEIP-S factors and the PSS.

## 3. Results

### 3.1. Descriptive Analysis

Table 2 shows descriptive statistics of the measured variables. Univariate normality using the Kolmogorov-Smirnov test and multivariate normality using Mardia’s test was not assumed (*p* > 0.05). Outliers’ analysis showed that atypical cases were not an important bias in the results because of the sample size.

### 3.2. Confirmatory Factor Analysis

The four models tested were: unifactorial, four correlated factors (default model), the bi-factor model, and the hieratic model. Given the non-normal distribution, the maximum likelihood method (MLM) was used. Table 3 shows the fit index for the four models analysed. The unifactorial model shows the poorest fit index, followed by the hieratic model. However, the four correlated factors model and the bifactor models showed similar and good values in the RMSEA, CFI, and TLI index. As shown in Table 4, only the unifactorial model showed statistically significant differences from the other three models. Given these results, and following the parsimony principle, the models proposed by the author, composed of four correlated factors, seem to be the most appropriate model to group our sample data. Following this model, Table 5 shows the factor structure of the four-factor models with a factorial load above 0.30.

### 3.3. Reliability Analysis and Nomological Validity

Table 6 shows the correlation coefficient between all the WEIP-S scales and the complementary psychological measures, and the alpha de Cronbach coefficient for all the factors of the WEIP-S. All the WEIP-S factors showed a significant direct association with the emotional clarity scale of the TMMS-24. However, no association was found between emotional repair and the management of their own emotions. Emotional attention was only found to be positively associated with the awareness of the other’s emotions. All the WEIP-S scales were negatively associated with the stress-perceived measure. Reliability analysis for the WEIP-S factors showed good internal consistency values for all factors, except for the management of one’s own emotions scale [28].

## 4. Discussion

The current investigation aimed to validate the factor structure of the WEIP-S in a sample of Spanish federated coaches. Likewise, to examine the psychometric characteristics of this tool. Such goals were driven mainly by the absence of specific surveys to assess EI in the field of sport. The confirmatory factor analysis showed that the four-factor model of WEIP-S was adequate for coaches. The reliability and convergent validity results were good in three out of four factors, except for ME. Given these results, the four-factor model is the most appropriate and parsimonious. These results are in line with previous studies that show a WEIP-S similar structure. These results were found in studies with workers’ samples [23,29], in the sport context with 273 athletes [20], and specifically with soccer players, the Portuguese WEIP-S version [30]. Likewise, these studies also show the ME factor to be the poorest in internal consistency. The two previous validation studies in the sports context [20,30] showed a similar internal consistency ME value to ours. Different authors establish 0.60 as the cut-off point [31,32]. However, the convergent validity value of ME leads us to consider some problems in the measurement of this factor. We consider that one of the possible causes may be the content of the items of this factor. Thus, only one of the items refers to one’s own emotions, and the rest are more related to conflict management with team members, which requires behavioural skills but not necessarily emotional management skills. For example, in item 8: "I listen impartially to the ideas of my team members.", coaches can respond to this item by considering what they would do in this situation, not what they would feel.

For the study of nomological validity, the correlations of the WEIP-S factors and the TMMS-24 dimensions were analysed. We found significant positive correlations between all WEIP-S factors and the subscales of the TMMS-24, except for emotional attention with the factors AE, ME, and MOE. These results are partially like Marchena-Giráldez et al. [20], who also found no correlation between the AE and ME factors. These results may highlight difficulties in measuring emotional attention as a linear variable [33]. According to the TMMS-24, individuals who score high on emotional attention pay too much attention to their emotions, which could lead to higher levels of anxiety and make it more difficult to manage emotions [18], as well as overreactions to negative emotions [34]. Furthermore, no correlation was found between the variable emotional repair and the ME factor, which may be due to the content of the items of this factor, as we have already mentioned. Regarding emotional clarity, every WEIP-S factor has a significant positive correlation. These results seem congruent since this dimension of the TMMS-24 focuses on understanding emotional states, and all WEIP-S items will score higher to the extent that emotional states are better understood.

Regarding the correlational analysis with the perceived stress variable, our study found significant inverse relationships with all the WEIP-S variables. The study conducted to validate the WEIP-S in athletes [20] found similar results for the variables ME and MEO. However, they did not find significant relationships between the variables AE and AEO with perceived stress. The PSS items refer to coping with stressful situations in the last month, where stress management requires better management of emotions to reduce stress [35]. The rest of the PSS tool refers to feelings derived from stressful situations. Coaches seem to perceive emotions derived from a stressful experience in the last month and can express and share them with the rest of the team, as well as recognise emotions in others, unlike athletes [20]. This may be due to the coach’s position and status, which obliges him to express his thoughts and feelings in talks with players and staff. For example, as Tamminen & Bennett [36] mention, the social and cultural context of sport justifies that the coach, because of his or her status, can express emotions such as anger. However, athletes are limited in their response options to the coach. Thus, some authors express the existence of social norms towards emotion in the sport context [37,38].

Even though many investigations have validated EI measurement instruments for the sports context [17], there was no evidence demonstrating the relationship between the emotional intelligence of coaches and how this affect their athletes.

However, there are limitations to our research that will need to be examined in further investigations. Firstly, the sample studied was not diverse regarding gender composition and the nature of the sport they coached. This limitation means that the findings on the factor structure of the WEIP-S in the field of sports should be treated with caution. Furthermore, in terms of reaching participants, snowball sampling was used. The main disadvantage of this, apart from having little control over the sample, is that the sample’s representativeness is not guaranteed. The actual distribution of the sample in the general population is not known. Lastly, related to the above, our survey is a cross-sectional study, so cause—effect relationships cannot be established.

Despite these limitations, our survey has several practical implications for sporting performance. It highlights the relevance of the rest of the team in the assessment of the coaches’ EI, an aspect generally missed in the measurement of this domain. Until now, the assessment of EI has focused only on the individual, suggesting that the whole EI construct is not being measured. According to this assumption, programs that aim to enhance sports performance using EI are unlikely to be addressing all dimensions of the domain, which may limit the efficacy of the interventions. Therefore, any future interventions developed to help people manage stress and enhance performance by improving EI in sports must consider these issues. To this end, it is essential to measure EI both at baseline and post-treatment, and WEIP-S appears to be an appropriate tool to measure all dimensions.

## 5. Conclusions

This study confirms that the WEIP-S questionnaire has good psychometric properties for measuring the coaches’ EI. The analyses conducted suggest that the four-factor model that considers the importance of awareness and management of emotions, both one’s own and those of others (players, coaching staff, etc.), is appropriate for measuring EI in this population. However, we found limitations with the reliability and convergent validity of the ME scale, as in previous studies.

The validation of this EI tool for sports coaches paves the way for the design of intervention studies to test whether emotionally intelligent coaches can manage teams and influence the EI of their players with greater probability than those with low levels of emotional awareness and management.

## Figures and Tables

**Table 1 ijerph-19-14371-t001:** Socio-demographic characteristics of the study.

Variables	N	%
Sex		
Male	95	59.0%
Female	66	41.0%
Worker		
Yes	153	95.0%
No	8	5.0%
Sport		
Rhythmic gymnastics	33	20.5%
Swimming	29	18.0%
Football	25	15.5%
Basketball	12	7.5%
Figure skating	12	7.5%
Water Polo	7	4.3%
Karate	7	4.3%
Hockey	7	4.3%
Skating	4	2.5%
Rugby	4	2.5%
Volleyball	4	2.5%
Handball	3	1.9%
Indoor Football	3	1.9%
Athletics	2	1.2%
Badminton	2	1.2%
Others	7	4.3%
Weekly hours of training		
Up to 2h	8	4.9%
2h to 7h	48	29.4%
7h to 14h	41	25.2%
14h to 21h	44	27%
Up to 21h	20	12.2%
Years of training		
Up to 1 year	3	1.9%
1 to 5 years	34	21.1%
5 to 10 years	39	24.2%
Up to 10 years	85	52.8%
Competition level		
Local	8	5.0%
Provincial	67	41.6%
National	77	47.8%
International	9	5.6%

**Table 2 ijerph-19-14371-t002:** Descriptive statistics for psychological tests.

	M	SD	Skewness (z-Score)	Kurtosis (z-Score)
WEIP-S: AE	22.32	4.48	−0.94	0.97
WEIP-S: ME	24.55	2.6	−1.00	1.83
WEIP-S: AOE	22.19	3.46	−0.95	1.55
WEIP-S: MOE	24.29	3.24	−1.54	3.79
TMMS: EA	27.66	5.82	0.11	−0.46
TMMS: EC	31.02	5.3	−0.21	−0.16
TMMS: ER	31.42	5.11	−0.72	0.83
PSS	22.82	8.19	0.19	−0.27

WEIP-S, Workgroup Emotional Intelligence Profile short version; TMMS, Trait-Meta Mood Scale; PSS, Perceived Stress Scale; AE, awareness of own emotions; ME, management of own emotions; AEO, awareness of other’s emotions; MOE, management of other’s emotions; EA, emotional attention; EC, emotional clarity; ER, emotional repair.

**Table 3 ijerph-19-14371-t003:** Fit index values for the WEIP-S models.

Model	Chi Square (df)	*p*	CFI	TLI	RMSEA (IC90)	SRMR	AIC	BIC
Unifactorial	321.1 (104)	<0.0001	0.638	0.582	0.142 (0.125–0.160)	0.115	7051	7048
Four-factors	100.8 (98)	0.404	0.995	0.994	0.017 (0.000–0.055)	0.056	6723	6720
Bifactorial	94.4 (88)	0.300	0.989	0.985	0.027 (0.000–0.061)	0.049	6730	6726
Hieratic	101.8 (100)	0.432	0.997	0.996	0.013 (0.000–0.054)	0.057	6721	6718

**Table 4 ijerph-19-14371-t004:** Model comparison.

Models	Δχ^2^	Δgl	*p*	ΔAICº	ΔBIC
Unifactorial/4 factors	220.3	6	<0.0001	328	328
Unifactorial/Bifactorial	226.7	16	<0.0001	321	322
Unifactorial-Hieratic	219.3	4	<0.0001	330	330
4-factors/Bifactorial	6.4	10	0.781	−7	−6
4-factors/Hieratic	1	2	0.638	2	2
Bifactorial/Hieratic	7.4	12	0.837	9	8

**Table 5 ijerph-19-14371-t005:** Factor loading of the four-factor models.

	AE	ME	AOE	MOE
WEIP 1	0.959			
WEIP 2	0.994			
WEIP 3	1.192			
WEIP 4	0.959			
WEIP 5		0.529		
WEIP 6		0.579		
WEIP 7		0.443		
WEIP 8		0.477		
WEIP 9			0.730	
WEIP 10			0.909	
WEIP 11			0.890	
WEIP 12			0.605	
WEIP 13				0.658
WEIP 14				0.679
WEIP 15				0.870
WEIP 16				0.825

WEIP-S, Workgroup Emotional Intelligence Profile short version; AE, awareness of own emotions; ME, management of own emotions; AOE, awareness of other’s emotions; MOE, management of other’s emotions.

**Table 6 ijerph-19-14371-t006:** Correlation coefficient and Cronbach’s alpha between all WEIP-S scales and the complementary psychological measures.

	TMMS: EA	TMMS: EC	TMMS: ER	PSS	α Cronbach
WEIP-S: AE	0.116	0.263 **	0.184 *	−0.199 *	0.841
WEIP-S: ME	0.065	0.252 **	0.118	−0.294 **	0.616
WEIP-S: AOE	0.193 *	0.382 **	0.191 *	−0.194 *	0.813
WEIP-S: MOE	0.096	0.232 **	0.291 **	−0.234 **	0.876

WEIP-S, Workgroup Emotional Intelligence Profile short version; TMMS, Trait-Meta Mood Scale; PSS, Perceived Stress Scale; AE, awareness of own emotions; ME, management of own emotions; AOE, awareness of other’s emotions; MOE, management of other’s emotions; EA, emotional attention; EC, emotional clarity; ER, emotional repair. * Significant at the 0.1 level. ** Significant at the 0.05 level.

## Data Availability

Any information is available to anyone who needs it upon justified request.

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
