# Peer review of "Psychometric Properties of the Spanish Version of the Work Group Emotional Intelligence Profile Short Version (WEIP-S) in a Sample of Spanish Federated Coaches"

_ijerph, 2022, doi:10.3390/ijerph192114371_

Round 1

Reviewer 1 Report

It was my pleasure to review this paper assessing the WEIP-S version performance in coaches. Authors have performed a well organized work. Sample size and statistical methods are properly presented. Results are clearly written. 

I have only some minor comments for the authors.

LINE 77: The first sentence ends writing and women. Is this a typo?

The characteristics of the included population - along with relevant table- should be transferred to the results section.

Please also provide the ethnicity as well as the education level of the participants if available.

Section 2.2. Please provide the questionnaire used as a supplementary file.

Could authors provide a subgroup analysis based on the type of sport, ie personal and team sports? Moreover a subgroup analysis based on the 

Author Response

Dear Reviewer, thanks for your comments and suggestions. I am sure that your reviews have helped improve the quality of our manuscript. Here is the answer to them:

  • LINE 77: The first sentence ends writing and women. Is this a typo? Thank you, we have corrected typographical error.
  • The characteristics of the included population - along with relevant table- should be transferred to the results section. Thank you for the suggestion. As we argued in your last comment, the objective of this study was not to analyze differences in emotional intelligence according to the characteristics of the sample. For this reason, the description of the sample has been included in the sample section instead of in the results.
  • Please also provide the ethnicity as well as the education level of the participants if available. We included ethnicity in the participant’s section. However, education level was not available.
  • Section 2.2. Please provide the questionnaire used as a supplementary file. The questionnaire is available to anyone who needs it in the references provided. However, it has been noted in the "Data availability statement" section that the information is available upon request.
  • Could authors provide a subgroup analysis based on the type of sport, ie personal and team sports? Moreover, a subgroup analysis based on the
    Thanks for the suggestion. We did not include these analyzes in the manuscript because they were not part of the study objectives. In addition, no statistically significant differences were found in emotional intelligence according to some variable as gender, hours of training, years of training or competition levels (p>.05). Other subgroups (e. g. type of sport) were not tested because of the lack of variability.

Reviewer 2 Report

I have decided to reject the article for lack of originality and little scientific contribution to the field of study. 

Author Response

Dear reviewer, thank you for your review and dedication. We consider that the study contributes by offering new evidence that the WEIP-S works in a sample of coaches. This finding allows us to advance in research so that now in the future intervention studies and improvement of emotional competencies can be proposed in all sports teams including staff, coaches and athletes.

Reviewer 3 Report

The paper presents a study that aimed to validate the Spanish version of the Work Group Emotional Intelligence Profile Short version (WEIP-S) in a sample of Spanish federated coaches. The paper's main contribution is the validation of WEIP-S to Spanish federated coaches.

The data are relevant to the field, and they are presented in a well-structured manner. However, the sampling procedure is not clearly described:

-        It is unclear if the research sample consisted only of coaches (men and women) or if there were also other women with different roles (see lines 77 and 78).  

-        The specification (in line 82) of the minimum number (150) of the sample should be explained, or the expression of this number should be rephrased/ reformulated concerning the research population.

-        The sampling procedure (snowball technique) should be argued, especially since it is a validation study and the sample should be representative of the professional category for which the psychological instrument is validated.

The paper offers figures/tables/images/schemes that properly show the data, so they are they easy to interpret and understand. Also, the data interpretation is appropriately and consistently throughout the manuscript, using adequate statistical analysis instruments. There the conclusions are consistent with the evidence presented.

Another issue needed to be solved to improve the clarity of the procedures is that authors should argue why they chose those very instruments, other than WEIP-S, presented in section 2.2. (Measures).

The ethics statements and data availability statements are adequate.

The cited references are mostly recent publications, but few of them are from within the last 5 years.

The paper can be published after the recommended modifications.

Author Response

Dear Reviewer, thanks for your comments and suggestions. I am sure that your reviews have helped improve the quality of our manuscript. Here is the answer to them:

  • It is unclear if the research sample consisted only of coaches (men and women) or if there were also other women with different roles (see lines 77 and 78).  Thank you, we have corrected the error. Thanks, we clarified this in the manuscript.
  • The specification (in line 82) of the minimum number (150) of the sample should be explained, or the expression of this number should be rephrased/ reformulated concerning the research population. We reformulated this expression to make the sample size estimation procedure more understandable.
  • The sampling procedure (snowball technique) should be argued, especially since it is a validation study and the sample should be representative of the professional category for which the psychological instrument is validated. We considered this to be the best method because the distinctive feature of the population we wanted to study tends to group these individuals together, to favor their social contact.
  • Another issue needed to be solved to improve the clarity of the procedures is that authors should argue why they chose those very instruments, other than WEIP-S, presented in section 2.2. (Measures). Measures section have been reformulated to clarify this aspect and to make them simpler and more understandable to the objectives.